

# A cross-sectional study on perceptions towards safe disposal of unused/expired medicines and its associated factors among the public in Saudi Arabia—a threat to the environment and health

Noohu Abdulla Khan[1,*], Vigneshwaran Easwaran[1,*], Khalid Orayj[1], Krishnaraju Venkatesan[2], Sirajudeen Shaik Alavudeen[1], Saad Ali Alhadeer[1], Abdulbari Ali Al Nazih[1], Ibrahim Hadi Saeed Al Afraa[1], Abubakr Taha Hussein[1], Sultan Mohammed Alshahrani[1], Mohammad Jaffar Sadiq Mantargi[3] and Sivakumar Vijayaraghavalu[4]

[1] Department of Clinical Pharmacy, College of Pharmacy, King Khalid University, Abha, Aseer region, Saudi Arabia
[2] Department of Pharmacology, College of Pharmacy, King Khalid University, Abha, Aseer region, Saudi Arabia
[3] Department of Pharmaceutical Sciences, Pharmacy Program, Batterjee Medical college, Jeddah, Saudi Arabia
[4] Department of Life Sciences (Zoology), Manipur University, Imphal, Manipur, India
[*] These authors contributed equally to this work.

Corresponding author
Vigneshwaran Easwaran,
vbagyalakshmi@kku.edu.sa

## ABSTRACT

**Background.** The unsafe disposal of pharmaceutical waste poses significant health hazards and causes environmental pollution on a global scale. The lack of specifically authorized guidelines in Saudi Arabia for the disposal of unused medicines available at home creates an undue economic burden and potentially threatens the environment and healthcare.

**Aim.** The current study aimed to determine the presence, disposal practices, and perceptions of unused or expired household medicines. Furthermore, it identifies the association between demographic characteristics and the presence, disposal practices, and perceptions of the safe disposal of unused/expired medicines. The study also intended to obtain opinions on methods to control the hazardous effects of waste medicines and promote awareness among the public about the safe disposal of unused/expired medicines.

**Methods.** This study is a web-based, cross-sectional questionnaire-based survey conducted in Saudi Arabia. The authors of the current study developed a questionnaire based on relevant literature. The study questionnaire comprises various domains such as demographic characteristics, presence and disposal of unused/expired medicines, perception of safe disposal of unused/expired medicines, and recommended improvement plan for safe disposal of used/expired medicines. Estimating internal consistency, expert review, and retranslation methods ensured reliability, face validity, and language validation. The results are expressed as frequency and percentages for categorical data. In addition, a chi-square test was also performed to find the association between the independent variables and the survey responses.
**Results**. Among the total population, 643 reported having unused/expired medicines at home, with antibiotics being the most common (79.4%). Symptom improvement is associated with accumulating unused medicines at home (71.7%). Age group, educational level, and occupational status were the predictors of the prevalence of waste medicines available at home ($p < 0.05$). The method selection for disposing of unused medicines was influenced by gender, age group, marital status, and educational level ($p < 0.05$), and the preferred method was putting them in the garbage (86.1%). Educational level is the most prominent factor associated with the perception of disposal of unused/expired medicines ($p < 0.001$).

**Conclusion**. Our study reveals a positive perception of the safe disposal of used or expired medicines, but practice requires improvement. The initiatives to improve the safe disposal practice should be tailored based on gender, educational level, and occupational status. Patient education during medicine dispensing could be an appropriate intervention and can be done by the pharmacist. Introducing medicine waste collection programs or safe medicine disposal guidelines for the public in Saudi Arabia could effectively prevent potential environmental and health hazards.

## INTRODUCTION

Unsafe medical disposal is a worldwide crisis because of its potential health hazards to the community and environment (*Letho et al., 2021*). Medical waste that is not adequately disposed of not only causes environmental pollution but also produces various infections (*Su, Wang & Li, 2021*). Recently, the emergence and outbreak of infectious diseases have placed significant strain on the healthcare system. During these outbreaks, medical waste displays new characteristics such as increased risk, continuous waste accumulation, and more rigorous disposal requirements (*Su, Wang & Li, 2021*). Poor management of medical waste causes significant health hazards to diverse populations, including healthcare professionals, waste workers, patients, *etc* (*Mohammed, Kahissay & Hailu, 2021*).

The World Health Organization (WHO) defines medical waste as the waste generated in the diagnosis, treatment or immunization of human beings or animals (*Su, Wang & Li, 2021*). Medical waste includes pharmaceutical waste as 'expired, unused, spilt and contaminated pharmaceutical products, prescribed and proprietary drugs, vaccines and serum that are no longer required and due to their chemical or biological nature, need to be disposed of carefully (*Chisholm et al., 2021*).

Pharmaceuticals are among the foremost contributing factors to environmental pollution, which poses significant risks due to their wide usage by various populations (*Vatovec et al., 2021*). Though different methods are suggested for disposing of pharmaceutical waste, insufficient information about pharmaceutical toxicity leads to unsafe pharmaceutical waste disposal, which compromises community safety and environmental pollution (*Desai, Njoku & Nimo-Sefah, 2022*). Various medications,

including antibiotics, beta-blockers, non-steroidal anti-inflammatory drugs (NSAIDs), *etc*, were found in multiple environmental compartments such as wastewater, surface water, groundwater, sediments, and soils (*Munzhelele et al., 2024*; *Eapen et al., 2024*). In addition, the disposal of unused/expired medicines through household waste damages the environment. It leads to toxic effects if taken by human beings, cattle, and other living things intentionally or accidentally (*Hassan, Taisan & Abualhommos, 2022*).

Immense pharmaceutical waste varieties were found in various countries' fresh and marine water. Thus, the presence of pharmaceuticals in aquatic environments has been a recent research topic (*Gworek et al., 2020*). Increased consumption of medicines worldwide for various medications, lack of policies, and inadequate awareness of environmental consequences are the reasons for the inflated accumulation of pharmaceutical wastes (*Alfian et al., 2021*; *Gubae et al., 2023*). The current practice of pharmaceutical disposal in various countries has brought a lot of emerging issues in recent years and is becoming a challenge for appropriate waste management (*Rogowska & Zimmermann, 2022*).

The Ministry of Health (MOH) is a significant healthcare provider in Saudi Arabia. The Saudi MOH passed regulations on healthcare waste management, which were subsequently endorsed by the Gulf Cooperation Council countries (GCC). The national policy on waste management was developed and revised in 2005 and 2019. In addition, the Saudi Waste Management Center was established in 2019 to organize and supervise waste management regulations and practices. However, the policy lacks adequate sustainable waste-handling guidelines (*Alharbi, Alhaji & Qattan, 2021*; *Gowdar et al., 2024*).

As reported in previous studies, Saudi Arabia has no specific authorized guidelines for disposing of unused or expired medicines available at home; however, the most frequently used method is to return the drugs to the pharmacy (*Alghadeer & Al-Arifi, 2021*). In addition, most Saudi families dispose of unused medicines into household wastes. In Gulf countries, including Saudi Arabia, pharmaceutical waste is introducing an additional healthcare cost to regular healthcare (*Althagafi et al., 2022*). Further, the number of research conducted concerning the safe disposal of medicines in the Middle East area is limited, particularly in Saudi Arabia (*Al-Shareef et al., 2016*; *Althagafi et al., 2022*).

Therefore, the current study was undertaken to estimate the presence, disposal practices, and perceptions of unused or expired household medicines. Furthermore, it explores the association between demographic characteristics and presence, disposal practices, and perceptions of safe disposal of unused/expired medicines. The study also intended to obtain opinions on methods to control the hazardous effects of waste medicines and promote awareness among the public about the safe disposal of unused/expired medicines. With these objectives, this study will help generate evidence regarding drug usage and disposal patterns among the general public living in Saudi Arabia. Understanding perception will also help identify gaps and barriers in the focus group and frame an appropriate interventional strategy to enhance community safety and decrease environmental pollution through pharmaceuticals.

## MATERIALS & METHODS

### Study design

It was a web-based cross-sectional observational survey conducted among the common public in Saudi Arabia. The study included the non-probabilistic convenience sampling technique, which targeted the citizens and residents of Saudi Arabia aged more than 18 years. The study was conducted between January 2021 and December 2021. The participants who were not willing to provide electronic consent to participate were excluded from the study. The incomplete responses to the questionnaire were excluded from the final analysis.

### Sample size

The sample size was calculated using Rao's soft sample size calculator. Considering the total number of people living in Saudi Arabia in 2019 (*General Authority for Statistics, 2019*), with a margin error of 5%, a confidence level of 95%, and a response distribution of 50%, the estimated sample size was calculated as a minimum of 385 participants.

### Study tool

The questionnaire was prepared to identify the perceptions and associated factors concerning disposing unused/expired medicines at home. The authors of the current study developed a questionnaire referring to relevant literature (*Al-Shareef et al., 2016*; *Ayele & Mamu, 2018*; *Gidey et al., 2020*; *Wajid et al., 2020*; *Azmi Hassali & Shakeel, 2020*).

Study tool description: The study questionnaire comprised 18 questions, including demographic characteristics (five questions), presence and disposal of unused/expired medicines (five questions), perception of safe disposal of unused/expired medicines (five questions), and recommended improvement plan for safe disposal of used/expired medicines (three questions). The study questionnaire comprised mixed questions such as dichotomous type, 4-point Likert scale, and open and closed-ended questions. The Likert scale used in the current study ranges from strongly agree to strongly disagree, where agree and strongly agree were considered positive responses, and disagree and strongly disagree were considered negative responses.

### Validity and reliability of the study tool

The questionnaire was initially prepared in English and translated into Arabic. The reverse translation from Arabic to English was done to guarantee the original meaning. However, the Arabic version of the questionnaire was circulated to get responses. The questionnaire was tested using a pilot survey conducted among 20 subjects, where the study participants were asked to complete the survey questionnaire. They also provided feedback at this point to improve the questionnaire. The results of the pilot study, as well as the suggestions from the feedback, were utilised to make necessary modifications to the questionnaire. The final version of the questionnaire was sent to experts in the field who are proficient in Arabic and English to ensure the questionnaire's face validity. The reliability was assured by calculating internal consistency; the estimated Cronbach $\alpha$ coefficient was 0.64. The results from the pilot study were not included in the derivation of the final results.

## Data collection

The finalized questionnaire was uploaded to Google Forms in Arabic. The Google Form link was distributed *via* various social media platforms like WhatsApp, Facebook, *etc.* The survey limited the multiple responses from a single participant by using the options available in the Google Form. The data collection period was January to December 2021. In order to administer the survey, the research assistants from various regions of Saudi Arabia reached out to the public, their relatives and their friends with an electronic device. They helped if they ran into problems or were illiterate. This procedure guaranteed both the proper completion of the survey questionnaire and the participation of respondents from across the nation.

## Ethical considerations

The electronic consent form was included at the start of the survey questionnaire, and only the agreed-upon subjects could complete it. The electronic consent form included the study objectives, study protocol, disclosure of information, voluntary nature of the decision, and the nature of the survey. This study was approved by the research ethics committee, King Khalid University ECM#2020-3305. Survey responses were analyzed aggregately to maintain data privacy and confidentiality. In addition, the survey was designed to avoid collecting potentially identifiable information. All the data were anonymized and used solely for research purposes.

## Statistical analysis

All data were coded and analyzed using the Statistical Package for Social Sciences or SPSS version 22.0 (IBM Corp., Armonk, NY, USA). Descriptive statistics were sub-grouped based on sociodemographic characteristics. Categorical data were summarized using frequency (n) and percentages (%). The current study deals with frequency distribution among various categories, such as disposal methods for used and expired medicines, frequency of positive and negative perceptions, *etc*, among survey respondents. Thus, the chi-square test was used to determine the association between the independent variables and the survey responses, and a $p$-value $\leq 0.05$ was considered statistically significant.

# RESULTS

## Demographic characteristics

The final analysis included 770 responses, of which 67% were from males. Most study participants were young adults aged 18 to 24 (40.1%), and only 1.2% of the current study comprised the geriatric population. Nearly 57.3% of the participants were single, and 76% had completed their college-level education. The student community occupies a significant portion of the current study population (41.3%). The complete participant details are found in Table 1.

## Presence of unused/expired medicines at home

Overall, 643 (83.5%) respondents reported having unused/expired medicines at home, including vitamin supplements, topical and oral antibiotics, *etc*, at the time of data

**Table 1  Demographic characteristics of the participants.**

| Characteristics | | Frequency | Percent |
|---|---|---|---|
| Gender | Male | 516 | 67 |
| | Female | 254 | 33 |
| Age | 18–24 years | 309 | 40.1 |
| | 25–39 years | 300 | 39 |
| | 40–59 years | 152 | 19.7 |
| | 60 years and above | 9 | 1.2 |
| Marital Status | Single | 441 | 57.3 |
| | Married | 329 | 42.7 |
| Educational level | Illiterate | 2 | 0.2 |
| | Secondary education | 183 | 23.8 |
| | College level education | 585 | 76 |
| Occupation | Self Employed | 88 | 11.4 |
| | Student | 318 | 41.3 |
| | Government Employee | 276 | 35.8 |
| | Housewife | 9 | 1.2 |
| | Others | 79 | 10.3 |

collection. The participants stated several reasons for having unused medicines, including the expiration date, stopping taking the medicines when feeling better, the doctor changing medication, *etc*. The details are given in Fig. 1.

## Association between demographic characteristics and the presence of unused/expired medicines at home

Table 2 describes the analysis of the association between various demographic characteristics and the presence of unused/expired medicines available at home. It reveals that symptom improvement is majorly associated with accumulating unused medicines at home (71.7%). The reasons stated by the study participants for the presence of unused medicines at home were not found to be significantly associated with any of the demographic characteristics included in the analysis. However, the age group ($p = 0.047$), educational level ($p = 0.019$), and occupation ($p = 0.004$) were significantly associated with unused/expired medicines' availability at home.

## Disposal practices of unused/expired medicines at home

Among our study participants, 24.8% believed they knew the appropriate method for safely disposing of unused/expired medicines. Many respondents reported disposing of more than two different pharmacological classifications of unused/expired medicines from their homes. Among those, oral antibiotics are the known pharmacological classification of medicines disposed of from home by the study participants (54.3%). The most common method selected for the disposal of unused medicines by the study participants was household trash (86.1%), followed by returning it to pharmacists (6.8%) and pouring it
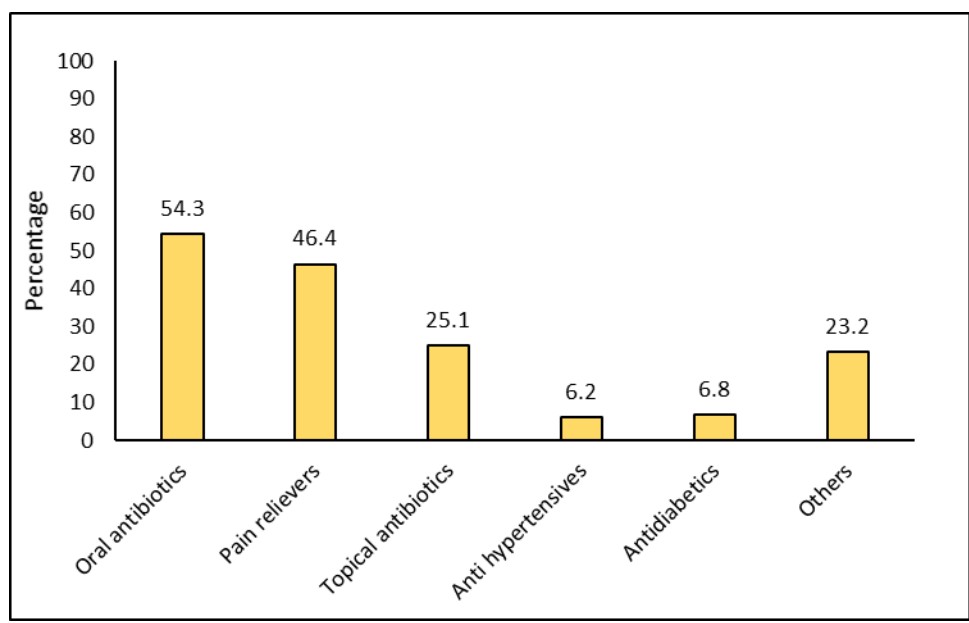

**Figure 1** Types of unused/expired medicines disposed of from home as reported by the study participants.

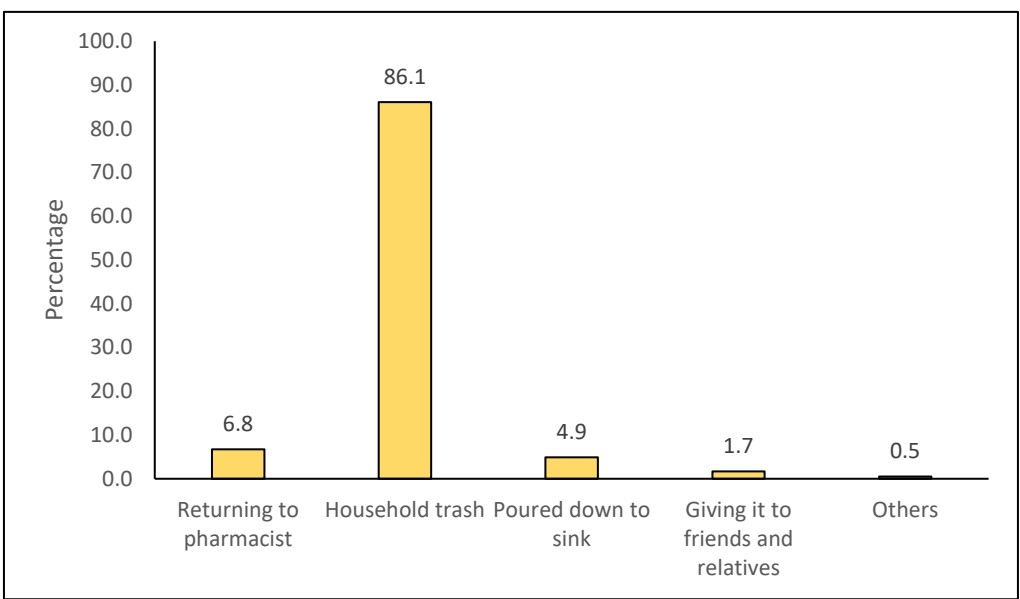

**Figure 2** Preferred method for the disposal of unused/expired medicines from home in our study participants.

down the sink (4.93%). The methods used by the public to dispose of unused or expired medicines are illustrated in Fig. 2.

**Table 2 Association between demographic characteristics and presence of unused/expired drugs at home.**

| Characteristics | Frequency of participants who had unused/expired medicines at home | p-value | Reasons | | | | | | | |
|---|---|---|---|---|---|---|---|---|---|---|
| | | | Medicine Expired | Symptoms improved, and I felt better | Doctor changed medication | Changed to herbal medicine | Did not feel a helping condition | Self-discontinuation | Others | p-value |
| **Gender** | | | | | | | | | | |
| Male | 427 | 0.421 | 21 | 315 | 24 | 1 | 16 | 43 | 7 | 0.465 |
| Female | 216 | | 15 | 146 | 17 | 1 | 9 | 27 | 1 | |
| **Age** | | | | | | | | | | |
| 18–24 years | 267 | | 12 | 194 | 18 | 1 | 6 | 31 | 5 | |
| 25–39 years | 253 | 0.047* | 18 | 179 | 14 | 1 | 11 | 28 | 2 | 0.064 |
| 40–59 years | 116 | | 6 | 85 | 7 | 0 | 7 | 11 | 0 | |
| 60 years and above | 7 | | 0 | 3 | 2 | 0 | 1 | 0 | 1 | |
| **Marital status** | | | | | | | | | | |
| Single | 378 | 0.056 | 22 | 269 | 25 | 2 | 12 | 43 | 5 | 0.803 |
| Married | 265 | | 14 | 192 | 16 | 0 | 13 | 27 | 3 | |
| **Educational level** | | | | | | | | | | |
| Illiterate | 1 | | 0 | 0 | 0 | 0 | 0 | 1 | 0 | |
| Secondary education | 142 | 0.019* | 8 | 101 | 14 | 0 | 6 | 11 | 2 | 0.312 |
| College level education | 500 | | 28 | 360 | 27 | 2 | 19 | 58 | 6 | |
| **Occupation** | | | | | | | | | | |
| Self employed | 75 | | 5 | 60 | 3 | 0 | 0 | 6 | 1 | |
| Student | 281 | | 18 | 196 | 21 | 2 | 9 | 29 | 6 | |
| Government employee | 222 | 0.004* | 8 | 166 | 13 | 0 | 12 | 23 | 0 | 0.228 |
| Housewife | 8 | | 0 | 4 | 1 | 0 | 0 | 3 | 0 | |
| Others | 57 | | 5 | 35 | 3 | 0 | 4 | 9 | 1 | |
| **Total** | 643/770 | | 36 | 461 | 41 | 2 | 25 | 70 | 8 | |

Notes.
*p-value less than 0.05, Chi-square test.

## Association between demographic characteristics and disposal practices for unused/expired medicines

Every demographic variable, except for occupational status, is associated with the study participants' choice about the disposal of unused or expired medicines. The complete details are provided in Table 3. Male gender, being single, and having a high level of education were the most prominent factors associated with the selection of methods for disposing of unused/expired medicines.

## Perception of safe disposal of unused/expired medicines

Many respondents in the current study had a positive perception of the safe disposal of unused or expired medicines (95.7%). The majority of participants in our study thought

**Table 3  Association between demographic characteristics and disposal practices for unused/expired medicines.**

| Characteristics | Returning to pharmacist | Household trash | Poured down to sink | Giving it to friends and relatives | Others | Total | *p*-value |
|---|---|---|---|---|---|---|---|
| **Gender** | | | | | | | |
| Male | 42 | 429 | 33 | 10 | 2 | 516 | 0.008[*] |
| Female | 10 | 234 | 5 | 3 | 2 | 254 | |
| **Age** | | | | | | | |
| 18–24 years | 9 | 276 | 18 | 3 | 3 | 309 | |
| 25–39 years | 22 | 260 | 12 | 5 | 1 | 300 | 0.004[*] |
| 40–59 years | 19 | 121 | 8 | 4 | 0 | 152 | |
| 60 years and above | 2 | 6 | 0 | 1 | 0 | 9 | |
| **Marital status** | | | | | | | |
| Single | 18 | 388 | 25 | 7 | 3 | 441 | 0.011[*] |
| Married | 34 | 275 | 13 | 6 | 1 | 329 | |
| **Educational level** | | | | | | | |
| Illiterate | 0 | 1 | 0 | 1 | 0 | 2 | |
| Secondary education | 20 | 150 | 6 | 6 | 1 | 183 | <0.001[*] |
| College level education | 32 | 512 | 32 | 6 | 3 | 585 | |
| **Occupation** | | | | | | | |
| Self Employed | 5 | 79 | 3 | 1 | 0 | 88 | |
| Student | 12 | 281 | 18 | 4 | 3 | 318 | |
| Government Employee | 31 | 226 | 13 | 6 | 0 | 276 | 0.166 |
| Housewife | 0 | 8 | 1 | 0 | 0 | 9 | |
| Others | 4 | 69 | 3 | 2 | 1 | 79 | |
| **Total** | 52 | 663 | 38 | 13 | 4 | 770 | |

Notes.
  *[*]p*-value less than 0.05, Chi-square test.

that unsafe disposal practices were a public health concern (96%) and that the public must be sensitized to their consequences. Considering the items included in the perception, few respondents (6.8%) believed that unused/expired medicines were not threatening the environment or community (6.4%). The extensive details are provided in Table 4.

### Association between demographic characteristics and the perception of safe disposal of unused/expired medicines

Table 5 describes the association between demographic characteristics and the perception of safe disposal of unused/expired medicines. All the demographic characteristics except age group impact item 1 (*i.e.,* Do you think unused/expired medicines present at home have potential risks for the community) Many male participants positively agreed that unused/expired medicines have potential risks ($p = 0.037$). Single subjects, students, and participants with higher education significantly differed from other subjects for item 1. They positively agreed that unused/expired medicines are a potential community risk.

Similarly, the age group of 25–39 years positively agreed with item 1. Though there is no significant difference between male and female participants, many male participants (62.8%) considered unsafe drug disposal as a threat to the environment

**Table 4  Frequency distribution of the public's perceptions towards of disposal of unused/expired medicines.**

| Items | Frequency of positive responses N (%) | Frequency of negative responses N (%) |
|---|---|---|
| **Item 1:** Do you think unused/expired medicines present at home have potential risks for the community? | 718 (93.2) | 52 (6.8) |
| **Item 2:** Do you think unsafe drug disposal practices threaten our environment? | 721 (93.6) | 49 (6.4) |
| **Item 3:** Do you think there is adequate information on the safe disposal of unused and expired household medicines? | 749 (97.3) | 21 (2.7) |
| **Item 4:** Do you think the public must be sensitized about the impact of unsafe disposal practices? | 758 (98.4) | 12 (1.6) |
| **Item 5:** Do you realize that the issue of medication disposal is of immediate public health importance? | 739 (96.0) | 31 (4.0) |
| **Total** | 737 (95.7) | 33 (4.3) |

($p = 0.793$). Similarly, single subjects and students agreed that unsafe drug-disposal practices threaten the environment, though they are statistically insignificant ($p = 0.563$ and 0.520, respectively). Participants with college-level education agreed that unsafe drug disposal practices are a threat to the environment, and it is statistically significant compared to another level of education ($p < 0.001$).

Well-educated subjects believed there was a lack of information on the safe disposal of unused/expired household medicines ($p < 0.001$). Similarly, they believed that the public must be alerted to the impact of unsafe disposal practices ($p < 0.001$). Young adults (38%), single subjects (54.55%), and participants with college-level education (73.11%) strongly believe that safe medication disposal is of immediate public health importance.

## Methods suggested by the public for environmental protection and awareness

The current study reports that 46% of the study participants prefer appropriate guidance regarding safely disposing of unused/expired medicines while purchasing medicines from the pharmacy. As reported by 40.1% of the study subjects, prescribing the medicines only in the required quantity will help reduce the hazardous effects of unused/expired medicines. In addition, 38.1% and 27.7% of the study participants stated that patient education by health care professionals and information *via* social media would help create awareness regarding the appropriate disposal of unused/expired medicines. The various methods suggested by the public for environmental protection and awareness are described in Table 6.

## DISCUSSION

Pharmaceutical waste management is still a significant challenge in the majority of countries. Identifying the issues related to pharmaceutical waste management and finding solutions for such community issues is our prime interest; therefore, we conducted this

**Table 5  Association between demographic characteristics and the perception of disposal of unused/expired medicines.**

| Characteristics | Item 1 | | Item 2 | | Item 3 | | Item 4 | | Item 5 | |
|---|---|---|---|---|---|---|---|---|---|---|
| | FPR[a] | p-value | FPR[a] | p-value | FPR[a] | p-value | FPR[a] | p-value | FPR[a] | p-value |
| **Gender** | | | | | | | | | | |
| Male | 488 | 0.037* | 484 | 0.793 | 502 | 0.973 | 510 | 0.206 | 496 | 0.763 |
| Female | 230 | | 237 | | 247 | | 248 | | 243 | |
| **Age** | | | | | | | | | | |
| 18–24 years | 278 | | 291 | | 297 | | 303 | | 293 | |
| 25–39 years | 283 | 0.545 | 277 | 0.128 | 294 | 0.088 | 296 | 0.263 | 289 | <0.001* |
| 40–59 years | 150 | | 145 | | 150 | | 151 | | 149 | |
| 60 years and above | 7 | | 8 | | 8 | | 8 | | 8 | |
| **Marital status** | | | | | | | | | | |
| Single | 397 | <0.001* | 411 | 0.563 | 425 | 0.507 | 433 | 0.229 | 420 | 0.004* |
| Married | 321 | | 310 | | 324 | | 325 | | 319 | |
| **Educational level** | | | | | | | | | | |
| Illiterate | 1 | | 1 | | 1 | | 1 | | 1 | |
| Secondary education | 164 | 0.027* | 169 | <0.001* | 175 | <0.001* | 180 | <0.001* | 175 | 0.012* |
| College level education | 553 | | 551 | | 573 | | 577 | | 563 | |
| **Occupation** | | | | | | | | | | |
| Self employed | 83 | | 80 | | 87 | | 85 | | 82 | |
| Student | 285 | | 302 | | 303 | | 311 | | 303 | |
| Government employee | 267 | 0.012* | 259 | 0.520 | 271 | 0.063 | 274 | 0.243 | 270 | 0.266 |
| Housewife | 9 | | 8 | | 9 | | 9 | | 9 | |
| Others | 74 | | 72 | | 79 | | 79 | | 75 | |
| **Total** | 718 | | 721 | | 749 | | 758 | | 739 | |

**Notes.**

Item 1: Do you think unused/expired medicines present at home have potential risks for the community? Item 2: Do you think unsafe drug disposal practices threaten our environment?, Item 3: Do you think there is adequate information on the safe disposal of unused and expired household medicines?, Item 4: Do you think the public must be sensitized about the impact of unsafe disposal practices?, Item 5: Do you realize that the issue of medication disposal is of immediate public health importance?

*p-value less than 0.05, Chi-square test.

[a]FPR-Frequency of Positive Responses.

research to understand the current level of awareness and trend of practice among the common public regarding pharmaceutical waste management in various regions of Saudi Arabia. Most of the study participants were young adults, and this might be because the university students are the initial point of contact for the current study. These results are similar to the study conducted in Johannesburg, South Africa (*Magagula, Rampedi & Yessoufou, 2022*).

## Presence of unused/expired medicines at home

Many of the respondents in our study reported that they disposed of more than two different categories of unused medicines from their homes. Among those, antibiotics are the most prevalent in our study, whereas painkillers are the predominantly available unused medicines from the study reported by *Magagula, Rampedi & Yessoufou (2022)*. However, a study by *Tegegne et al. (2024)* found that antibiotics were the most common type of medicine present unused at home. Next to antibiotics, painkillers are found to be present

**Table 6  Methods suggested by the public for environmental protection and awareness.**

| Type of methods | Frequency | Percent |
|---|---|---|
| *Methods to control the hazardous effects of unused/expired medicines* | | |
| Providing proper guidance to the consumer | 354 | 46.0 |
| Reducing the number of prescribed medications | 54 | 7.0 |
| Prescribe in specific quantities for the duration of treatment | 309 | 40.1 |
| Donating or sharing unused medication | 49 | 6.4 |
| Others | 4 | 0.5 |
| *Methods to create awareness among the community towards safe disposal of drug* | | |
| Awareness program by the government | 84 | 10.9 |
| Patient education by HCP | 293 | 38.0 |
| Written instruction on medicines | 180 | 23.4 |
| Information via social media | 213 | 27.7 |
| *Preferred mode of information* | | |
| Through healthcare professionals | 385 | 50.0 |
| Through Social media | 375 | 48.7 |
| Others | 10 | 1.3 |

unused at home; we observed similar results in a study conducted in the Jeddah population (*Abuassonon et al., 2019*). The primary reason stated by the current study subjects for having such medicines in households was to stop using the drugs once they felt better or no longer suffered from the condition. These findings are consistent with previous studies conducted in various parts of the world (*Wieczorkiewicz, Kassamali & Danziger, 2013*; *Marwa et al., 2021*; *Tegegne et al., 2024*). As reported by previous studies, improvement in symptoms leads to medication nonadherence. Thus, medication non-adherence and not completing the whole regimen due to the resolved symptoms led to the availability of unused medicines at home (*Rajavardhana et al., 2016*; *Insani et al., 2020*). It was suggested by *Kahsay et al. (2020)* that medicines should be prescribed in correct quantities to the patients and that patients should be encouraged to complete the whole treatment regimen even if the symptoms get better to contain the accumulation of pharmaceutical wastes. Additionally, as a recommendation to the pharmacists, the dispensing of unnecessary medicines should be avoided, and the medicines should be dispensed only for the required duration of treatment or as written in the prescription.

## Association between demographic characteristics and the presence of unused/expired medicines at home

Age group is one of the major factors associated with the availability of unused/expired medicines at home. These findings align with the study reported from Malaysia, which reported that unused medicines' availability decreases as age increases (*Wang, Aziz & Chik, 2021*). There is an exciting finding in the current study that the higher the education, the more wasted medicines are available at home; in this regard, multiple reasons have been stated by the study participants. This implies that the educated population lacks awareness

of appropriate self-health management and relies more heavily on medicines than other health measures, possibly accumulating medicine wastage. These results align with the study conducted among Malaysian residents (*Wang, Aziz & Chik, 2021*). Despite the fact that education is a significant factor that tends to influence the availability of unused or expired medicines, people of all educational backgrounds were generally willing to reuse unused stored medications (*Alhamad et al., 2023*). However, it is not applicable for expired medicines, as they cannot be reused. Most of the demographic characteristics were not found to influence the reasons stated by the study participants for having wasted medicines. Like our study, medicine storage behavior and the availability of unused medicines are heavily influenced by occupational or employment status, as reported in a study from Ethiopia (*Yimer, Moges & Kahissay, 2024*).

## Disposal practices of unused/expired medicines at home

In addition, improper disposal and lack of awareness of health risks associated with antibiotic exposure to environmental use are the major causes of environmental pollution (*Karimi et al., 2022*). These results align with our study, as most of our study participants disposed of their unused/expired medicines in the household trash. In our study, the participants preferred appropriate guidance from the pharmacy personnel when purchasing the drug. In this regard, pharmacists hold the major responsibility to make the patient utilize the medications responsibly. Our literature survey revealed similar results and opinions from other studies conducted in the UK and Kabul (*Bashaar et al., 2017*; *Watkins et al., 2022*). Household garbage is the most frequently opted method for disposing of unused medicine by the current study subjects. It seems to be the regular method adopted by the common public for disposing of unused/expired medication across various countries (*Sonowal et al., 2016*; *Hassan, Taisan & Abualhommos, 2022*). Besides, returning unused/expired medicines to the pharmacist was frequently reported in multiple studies (*Al-Shareef et al., 2016*); however, this habit is infrequently observed among the current study subjects and tends to be influenced by numerous factors similar to other studies (*Watkins et al., 2022*).

## Association between demographic characteristics and the disposal practices of unused/expired medicines at home

A study from Pakistan reported that female subjects dispose of unused medicines in the household garbage, and male subjects toss them on the ground (*Shah et al., 2023*). Similarly, age, gender, marital status, and education significantly affect the disposal practices of unused/expired medicines, which is similar to our study results (*Hajj et al., 2022*). As an additional fact, gender not only influences the disposal practice of pharmaceuticals but also includes used face masks, which are considered medical waste similar to pharmaceutical waste (*Easwaran et al., 2024*). The Food and Drug Administration (FDA) recommends drug take-back programs, returning to a nearby pharmacy, and appropriate household disposal methods as the best disposal method for unused/expired medicines (*Woldeyohanins et al., 2021*). However, it is evident that some of these methods are not available in some countries, and the public does not know about some (*Rogowska & Zimmermann, 2022*).

### Perception towards the safe disposal of unused/expired medicines

Our study respondents had a positive perception, similar to the study conducted among the public from Libya (*Alssageer, 2024*). Similar to the current study respondents, pharmacists also believe that improper disposal of unused medicines is a significant environmental threat (*Alfian et al., 2023*). Though the current study results reveal that the study participants had enough information, the availability of unused medicines is still prominent, and the methods selected for the disposal of unused medicines are inappropriate among many respondents, which indicates the requirement for increasing awareness and effective education (*Jankie et al., 2022*; *Alssageer, 2024*). Education by hospital and community pharmacists to the patient is considered the best option to fill the gap between awareness and perception and to practice the safe disposal of unused or expired medicines. However, pharmacists may require support from the government, standard guidelines, appropriate training, *etc* (*Alghadeer & Al-Arifi, 2021*; *Abbas et al., 2025*). The positive perceptions found in this study align with research done in Ethiopia that demonstrated that inappropriate disposal of used or expired medicines is a major public health issue (*Woldeyohanins et al., 2021*). This positive perception can be changed into appropriate practice with the collaboration of health care practitioners, including pharmacists.

### Association between demographic characteristics and the perception of safe disposal of unused/expired medicines

The common public believed unused/expired household medicines could cause environmental and public health problems (*Gidey et al., 2020*). Male participants in our study significantly differed from female participants in believing that unsafe drug disposal is a threat to the environment. Likewise, single subjects and participants with higher education also significantly believed that unsafe drug-disposal practices risk the environment. These findings align with similar studies conducted among pharmacy and nursing students from Kosava (*Shuleta-Qehaja & Kelmendi, 2022*). Students and well-educated subjects from the current study significantly understand the detrimental effects of the unsafe disposal of unused/expired medicines compared with other occupations and educational levels, respectively. However, safe practices are not being followed. The reason might be a lack of awareness and education about the method of disposal (*Shuleta-Qehaja & Kelmendi, 2022*). Eventually, the educated subjects from our study indicated a lack of information regarding the safe disposal of unused/expired medicines. As a result, they emphasized the need to raise awareness and provide quality information to alleviate public health concerns associated with waste medicines. The Malaysian study found that knowledge and awareness among the public were adequate, but responsible medication use was poor. Therefore, they suggested that healthcare professionals should be involved in educating patients (*Ling et al., 2024*), like the preferences stated by our study participants.

### Methods suggested by the public for environmental protection and awareness

The current study results show that participants received less information about the safe disposal of unused/expired medicines. The participants also insisted on various methods to improve public awareness regarding the hazards associated with unsafe disposal and the

safe disposal of unused or expired medicines. Most of the current study subjects preferred guidance on the appropriate disposal of medicines, and it was recommended that they be given only for the required duration of treatment to avoid wastage. These results are also aligned with those reported from Ethiopia and Saudi Arabia (*Kahsay et al., 2020*; *Althagafi et al., 2022*). Similar to our study, a study among pharmacy students revealed that patient counseling is an effective way to improve the awareness of the safe disposal of unused or expired medicines (*Shakib et al., 2022*).

### Strengths and limitations

One strength of the current study is its appropriate sample size. The study also collected samples from all over Saudi Arabia, representing all major regions of the country. Appropriate background search and data analysis strengthen the study. However, the current study has a huge amount of student participation, which limits the generalizability of the results to the whole of Saudi Arabia, which has varied population groups. In addition, the convenience sampling used in our study also limits the generalizability of the results. The current study focuses on perceptions; thus, future studies are warranted to estimate the practice at all levels, including the public, health care professionals, and health care settings. Recalling bias is possible as the current study responses are based on past events from memory. Response bias is possible when considering the mode of data collection. Direct interviews based on home visits are suggested for future research to reduce response bias.

## CONCLUSIONS

In conclusion, our study reveals a positive perception of the safe disposal of used or expired medicines. However, our study demonstrates that many unused medicines were found in Saudi community homes, and most pharmaceutical waste is disposed of in standard household garbage. Unsafe drug disposal and the high number of unused antibiotics available at home threaten the environment and may lead to antibiotic resistance in the future. Therefore, age group, educational level, and occupational status should all be considered when designing tailored interventions.

Symptom improvement was found to be the primary reason associated with the availability of unused medicines. This indicates that medicine users, preferably the young population, must be educated about the appropriate completion of prescribed medicines. The pharmacist should disseminate this information from the pharmacy while dispensing medicines.

The knowledge and awareness of the young population were found to be adequate. However, the practice is inappropriate. Thus, the young population, ideally the students, should be focused on initiatives that can help improve the practice.

More specifically, it is recommended that patient education and guidance about the safe disposal of unused/expired medicines at the point of medicine collection be provided since this is the most preferred method by the public.

The lack of medical waste collection methods from the public is a significant barrier that hinders the safe disposal of unused/expired. Therefore, introducing a pharmaceutical

waste collection program in public gathering points such as airports and shopping malls can bring a new vision and help solve the current issue. As per Vision-2030, one of the vision statements is "at the heart of our vision is a society in which all enjoy a good quality of life, a healthy lifestyle, and an attractive living environment," so this study results will explore new ideas and fill the gaps in pharmaceutical waste management and protect the environment in future. From this perspective, our study can bring a new achievement for the Kingdom of Saudi Arabia that aligns with Vision 2030.

## ACKNOWLEDGEMENTS

All the authors are highly thankful to those directly or indirectly involved in completing the study.

### Funding
This research was supported by the Deanship of Scientific Research and Graduate Studies at King Khalid University through a large group Research Project under grant number RGP2/330/45. The funders had no role in study design, data collection and analysis, decision to publish, or preparation of the manuscript.

### Grant Disclosures
The following grant information was disclosed by the authors:
The Deanship of Scientific Research and Graduate Studies at King Khalid University through a large group Research Project: RGP2/330/45.

### Competing Interests
The authors declare there are no competing interests.

### Author Contributions
- Noohu Abdulla Khan conceived and designed the experiments, authored or reviewed drafts of the article, and approved the final draft.
- Vigneshwaran Easwaran conceived and designed the experiments, performed the experiments, analyzed the data, authored or reviewed drafts of the article, and approved the final draft.
- Khalid Orayj conceived and designed the experiments, authored or reviewed drafts of the article, and approved the final draft.
- Krishnaraju Venkatesan performed the experiments, authored or reviewed drafts of the article, and approved the final draft.
- Sirajudeen Shaik Alavudeen performed the experiments, analyzed the data, prepared figures and/or tables, and approved the final draft.
- Saad Ali Alhadeer performed the experiments, prepared figures and/or tables, authored or reviewed drafts of the article, and approved the final draft.
- Abdulbari Ali Al Nazih performed the experiments, prepared figures and/or tables, authored or reviewed drafts of the article, and approved the final draft.

Peer J

- Ibrahim Hadi Saeed Al Afraa performed the experiments, authored or reviewed drafts of the article, and approved the final draft.
- Abubakr Taha Hussein conceived and designed the experiments, prepared figures and/or tables, and approved the final draft.
- Sultan Mohammed Alshahrani conceived and designed the experiments, analyzed the data, prepared figures and/or tables, authored or reviewed drafts of the article, and approved the final draft.
- Mohammad Jaffar Sadiq Mantargi conceived and designed the experiments, analyzed the data, prepared figures and/or tables, authored or reviewed drafts of the article, and approved the final draft.
- Sivakumar Vijayaraghavalu conceived and designed the experiments, prepared figures and/or tables, authored or reviewed drafts of the article, and approved the final draft.

## Human Ethics

The following information was supplied relating to ethical approvals (i.e., approving body and any reference numbers):

This study was approved by the research ethics committee, King Khalid University, ECM#2020-3305.

## Data Availability

The raw data is available in the Supplemental File.

## Supplemental Information

Supplemental information for this article can be found online at http://dx.doi.org/10.7717/peerj.19258#supplemental-information.

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
