# Peer review of "A cross-sectional study on perceptions towards safe disposal of unused/expired medicines and its associated factors among the public in Saudi Arabia—a threat to the environment and health"

_PeerJ, doi:10.7717/peerj.19258_

## Round 0.1 · original submission · Major Revisions

This study addresses major health issues and environmental contamination by revealing medical waste disposal procedures and public awareness. Basic Reporting, Experimental Design, and Finding Validity have been suggested for improvement. Congratulations to the authors, and with the suggested changes, the manuscript can contribute to the literature.

Reviewer 1 ·

Basic reporting

No comments

Experimental design

1. Line 47, Aim: The aim of the study is not aligned with the title. While the title suggests that the study focuses on factors influencing the safe disposal of unused or expired medicines, the aim states that the study is to evaluate public awareness. I encourage the author to review and adjust the aim to ensure it is consistent with the study title. This will ensure coherence between the aim and the intended outcome of the study

Validity of the findings

The conclusion is not aligned with the aims of the study and the study outcome overall

Additional comments

3. Lines 113 to 114: As mentioned earlier, it is necessary to align the study outcome with the study's aim. Currently, it is unclear whether the outcome focuses on public awareness or the factors influencing safe disposal

4. The sentence should read “ This was a web-based cross-sectional observational survey”

5. Line 127: Knowledge has been introduced as an additional aim alongside awareness. This needs to be clarified and addressed to maintain consistency with the study's primary objectives

6. Was the questionnaire developed or adapted? If it was developed based on relevant literature, appropriate citations should be included line 128-129

7. The questionnaire seem not to capture questions awareness, knowledge and factors influencing the safe disposal of unused or expired medicines. The tool captures perception which again is not part of the study aim or outcome.

8. The questionnaire does not appear to include questions on awareness, knowledge, or factors influencing the safe disposal of unused or expired medicines. Instead, it focuses on perceptions, which are not aligned with the study's aim or intended outcomes. The author needs to address this discrepancy ( line 127-134)

9. There appear to be no results reported on the knowledge, awareness, and factors influencing the disposal of unused or expired medicines. As it stands, the study's outcome remains unclear since only the results regarding perceptions were presented.

Annotated reviews are not available for download in order to protect the identity of reviewers who chose to remain anonymous.

·

Basic reporting

a. Proofreading is required to ensure the appropriate use of punctuation when writing. Some of the punctuations are in lines 77 and 80. The Author should use P<001 for the P-value of P=0.000. The total percentage at the educational level (Table 1) is not 100.0%
b. The Author wrote an appropriate background but needs to be more careful in reference utilization. Some references published more than 10 years ago are used in the introduction. The use of recent research results in the introduction is encouraged. Line 249-251 stated, "These findings are consistent with those documented in previous studies from the USA, Africa, and other countries (Wieczorkiewicz, Kassamali & Danziger, 2013; Marwa et al., 2021). The study covers only two cities but supports a statement covering many countries.
c. The Author wrote a well-structured manuscript.
d. No figure is used. Raw data is well supplied.

Experimental design

a. The manuscript meets the original primary research requirement within the journal's Scope.
b. Overall, the research question is well-defined, relevant & meaningful. It is stated how the research fills an identified knowledge gap. To provide a more precise description, the Author can make some improvements related to the topic, including:
1. Abstract. It is mentioned, "Aim: Therefore, it is necessary to critically evaluate public awareness regarding domestically disposing of unused/expired medicines and develop a plan to address it.” The Author can rephrase the sentences to describe the research objectives better because the research methods do not provide enough support for developing a plan as a research outcome. The methods section should describe the research method in more detail. The discussion of research instruments dominates the method, so many significant aspects of the method are not conveyed. The number of participants mentioned in the method and result sections is inconsistent, and there is no explanation for this inconsistency. The results and conclusions should match the research objectives.
2. Introduction. The Author implicitly describes the knowledge gap in the introduction. As stated in lines 97-102, the Author mentions several studies that reported on the disposal of unused/expired drugs in Saudi Arabia but has not explained the gaps this study will fill. The Author can mention the limitations of previous studies and improvements applied in this study to explicitly describe the knowledge gap.
c. Generally, research shows rigorous investigation performed with sufficient technical & ethical standards, but some improvements can be made in some aspects, including :
1. Study tool. The Author needs to mention the reference used to develop the questionnaire and explain how to ensure the face validity of the questionnaire so that the questionnaire can function according to the expected purpose, not be ambiguous, and cause misinterpretation.
2. Author need to add appropriate scientific background to ensure that the sample used adequately represents the population reached by the study (e.g., sample size calculation). In addition, the Author also needs to explain the method used to ensure that respondents represent various regions of Saudi Arabia, as stated in lines 234-236: "Therefore, we conducted this research to understand the current level of awareness and trend of practice among the common public regarding pharmaceutical waste management in various regions of Saudi Arabia. The author must explain how to ensure that illiterate respondents (Table 1) answer according to their opinions. Since the questionnaire was distributed online and in written form, Illiterate respondents needed someone else to help them read and answer it.
3. Lines 162-163 state, "while continuous data were expressed as the mean and standard deviation (±SD)." However, the reviewer did not find any data presented in mean and SD. In addition, the researcher used the Likert scale instead of the dichotomous scale but gave binary coding (Line 134-137; supplementary file) so that it did not produce ordinal data according to the data commonly generated by the Likert scale. Authors need to explain the reasons for the inconsistencies that occur. The author can explain if there is a change in the research method and the reason for it.
4. The Author needs to refer to the data analysis that supports the statement in the line, "It is also evident from the results that students and well-educated subjects tend to have many unused medications for varied reasons." (lines 185-186)
5. The author needs to refer to the data analysis that supports the statement in lines 271-275: "We found a lack of awareness as an imperative issue among the public towards safe disposal methods for unused/expired medicines; similar results were found in other studies (Corcoran, Winter & Tyler, 2010; Lubick, 2010)."
6. The author needs to refer to the data analysis that supports the statement in the line Line 309-310: "Most of our study's respondents showed interest in learning safe medication disposal methods."
7. Lines 190-191 state, "Many respondents reported disposing of more than two classifications of unused/expired medications from their homes." The standards or references used to classify the unused/expired medications and how to ensure respondents understand how to classify the medications they dispose of needs to be explained. Some drugs have multiple indications; for example, ibuprofen can be used as an analgesic (pain reliever) and antipyretic (other).
8. The strengths and limitations of the study need to be added

Validity of the findings

a. The study has a meaningful impact.
b. All underlying data have been provided; they are robust, statistically sound, & controlled.
c. Conclusions must be in line with the study objective. The conclusion needs to be improved to be well stated, linked to the original research question & limited to supporting results. The author can start by consistently addressing the study objective, as mentioned in some sections of the manuscript: “Aim: Therefore, it is necessary to critically evaluate public awareness regarding domestically disposing of unused/expired medicines and develop a plan to address it (line 47-48)” and “This study will help generate evidence regarding drug usage and disposal patterns among the general public living in five major regions of Saudi Arabia. It will also help identify gaps and barriers in the focus group and frame an appropriate interventional strategy to increase the community's safety and decrease environmental pollution through pharmaceuticals (lines 109-113)." The Author also needs to mention the vision stated in the conclusion. “..... and reach new achievements for the Kingdom of Saudi Arabia in alignment with Vision 2030”( lines 324-325).

Additional comments

Dear Authors,
Congratulations on your work!
The Author has produced significant work on drug safety that considers environmental concerns.

Reviewer 3 ·

Basic reporting

1. The manuscript is generally well-written, with clear and professional language used throughout. However, please ensure that terminology used throughout the manuscript is consistent, particularly when referring to “medical waste” and “pharmaceutical waste.” It would be beneficial to clarify these terms early in the manuscript to avoid confusion. Minor typographical errors were also found, such as the incorrect placement of punctuation before citations (e.g., lines 77 and 80). It is also suggested that table references be placed inside parentheses, for example “(Table 1).” Please conduct a thorough check for typographical errors throughout the manuscript to ensure consistency and accuracy.

2. The title does not accurately reflect the study design. It is suggested to revise the title to better represent the nature of the study, including the design and key focus. For example, adding a phrase such as “A Cross-Sectional Study on…” would provide a clearer indication of the research method.

3. The introduction provides sufficient context and references relevant literature. However, there are opportunities to improve the clarity and arrangement of the paragraphs to enhance the logical flow of the background section.
a. It is recommended to elaborate more on the hazards of unsafe medical disposal in the introduction to strengthen the rationale for conducting the study.

b. Consider merging the second sentence through the last sentence of the first paragraph (lines 75-80) with the second paragraph, and improving the paragraph to provide a more concise and coherent explanation of medicine waste as a key contributor to environmental pollution.

c. Additionally, a paragraph discussing the causes of the phenomenon (e.g., current practices of medicine disposal, particularly in Saudi Arabia) would improve the comprehensiveness of the background section.

d. Some of the cited literature is not recent (lines 89-90 and 92), and it is suggested to reference more recent studies to support the timeliness of the research.

4. The overall structure of the manuscript adheres to PeerJ standards and the norms of the discipline. The sections are appropriately divided, and the flow of information is logical.

5. The manuscript does not contain any figures. It is recommended to replace Table 3 and Table 6 with figures for better visualization of the data. This would enhance readability and improve the overall presentation of the results.

6. The raw data has been provided, which complies with PeerJ’s data availability policy.

Experimental design

1. The manuscript presents original primary research that falls within the scope of the journal. The study addresses an important issue regarding medical waste disposal practices.

2. The research question is well defined, relevant, and meaningful. It is clearly stated how the study fills an identified knowledge gap. However, the aim of the study should be stated consistently in both the Abstract and Introduction to ensure clarity and consistency. Furthermore, since this is an observational study, it would be more appropriate to use the term “identify factors associated with” rather than “evaluate the influence of” in describing the study's objectives.

3. The study generally meets the criteria for rigorous investigation in terms of its technical and ethical standards. The research design is appropriate, the variables are clearly defined, and the data collection process appears to be conducted consistently and reliably. The study obtained ethical approval from the relevant ethics committee, and informed consent was obtained from all participants. However, the manuscript does not clearly state how participants' data privacy and confidentiality were maintained. It is recommended to explicitly mention that data were anonymized and used solely for research purposes to ensure compliance with ethical standards.

4. The methods are generally well described, but it would be beneficial to add more details:
a. Please specify the minimum required sample size and provide a description of how it was calculated, referencing appropriate literature to support the calculation.

b. The authors mention that a questionnaire was developed based on relevant literature (lines 128-129), but the specific references used to develop the questionnaire are not provided. It is recommended to include these references to strengthen the validity of the questionnaire.

c. The authors mention conducting a pilot study with 20 participants (line 140). However, it is important to include a reference supporting the use of 20 participants for the pilot study.

d. Additionally, it is suggested to calculate item correlations (Pearson’s correlation) to assess the validity of the questionnaire items, as Cronbach’s alpha primarily measures reliability rather than validity. Reporting both validity and reliability values would enhance the credibility of the instrument.

e. Please provide more information on how the study prevented multiple participation from the same respondents to avoid duplication of data.

f. The manuscript lacks details on inferential statistics used to assess the relationships between variables (line 163). Please include information on the inferential statistical methods employed.

g. It would be more useful to categorize the levels of awareness and identify the associated factors using multivariate analysis, considering that these factors naturally occur together. This approach would also strengthen the findings by confirming whether these factors remain significant after controlling for potential confounding variables.

h. In the Abstract section, it is recommended to add information about the validity and reliability of the questionnaire. Providing details on the domains of the questionnaire would be more useful than simply describing the types of questions used (e.g., dichotomous, Likert scale, open-ended). Please also include details on the statistical analysis methods and software used in the study to ensure clarity.

Validity of the findings

1. The underlying data appear to be robust and well-controlled. The authors are commended for providing comprehensive data and ensuring transparency in their analysis. However, improvements are needed in the reporting and analysis of the data, particularly in the presentation of the sample size, as suggested in the Experimental Design section.

2. The authors report that a total of 871 responses were received, but only 770 responses were included in the final analysis after applying the exclusion criteria (lines 169-170). However, if the 871 responses include those who did not consent to participate, it would be more appropriate to only report the number of participants who provided consent (770). Reporting the initial 871 responses could give the impression that a large portion of the data was lost, which may not be the case.

3. Given that the main research question involves assessing the level of awareness, it is recommended that the manuscript present these data first before discussing the associated factors. This would improve the clarity and logical flow of the results section.

4. The Discussion section would also benefit from starting with a statement of the main findings before describing the associated factors. Structuring the discussion according to the research questions would improve clarity. For example, begin with a discussion of the level of awareness, followed by an analysis of the associated factors.

5. There are statements in the Discussion section that require clarification and referencing. For example, in line 243, the phrase “these results contradict” should be followed by a clear statement of what specific medicine is being referred to. In lines 307-309, the sentence “Our literature survey ...” is not referenced. It is recommended to provide appropriate references to support this statement.

6. The manuscript should also include a limitations section and recommendations for future research and practice to provide a balanced interpretation of the findings. Additionally, it is recommended to discuss the generalizability of the findings to ensure readers understand the broader applicability of the results.

7. The conclusions are generally well stated, but they do not fully answer the main research question. The conclusion section should start by answering the research question, for example, by categorizing the level of awareness (e.g., high, moderate, or low) and identifying the factors associated with these levels. Once the main research question is addressed, the conclusion can then move on to provide recommendations or implications for future research and practice. It is important to ensure that the conclusion remains focused on the study’s findings.

8. The same issue is observed in the Abstract, where the conclusion section does not directly address the research question. It is recommended to revise the abstract’s conclusion to clearly state the findings related to the level of awareness and associated factors.

Additional comments

This paper contributes valuable insights into medical waste disposal practices and public awareness levels, addressing significant health issues and the growing concern over environmental pollution. Suggestions for improvement have been provided in three core areas (Basic Reporting, Experimental Design, and Validity of Findings). The authors are commended for their efforts, and with the recommended revisions, the manuscript can make a meaningful contribution to the literature.

---

## Round 0.2 · Minor Revisions

The authors improved clarity, structure, and methodological rigor. To improve the work, the discussion and conclusion should properly synthesize the connected elements and state the significant findings. Most of the modifications are well-executed, and they will polish the work of writing. However, minor revisions need before publication in PeerJ.

·

Basic reporting

a. Proofreading is still required to ensure the appropriate use of punctuation when writing. Some of the punctuations are in lines 9, 208, and 271. The Author should use P<001 for the P-value of P=0.000 (the meaning is different). The total percentage at the “Methods to create awareness among the community towards safe disposal of drug” (Table 6) is not 100.0%. The number at the beginning of the sentence should be changed to the word (Line 236-237). Statistical Package for Sciences should be Statistical Package for the Social Sciences (SPSS) or Statistical Product and Service Solutions
b. The Author wrote a well-structured manuscript.
c. Figures are good
d. Raw data is well supplied.

Experimental design

a. The manuscript meets the original primary research requirement within the journal's Scope.
b. Overall, the research question is well-defined, relevant, and meaningful. It is stated how the research fills an identified knowledge gap.
c. Generally, research shows rigorous investigation performed with sufficient technical and ethical standards, but some improvements still can be made in some aspects, including :
1. Study tool. The Author needs to explain in more detail the purpose of distributing questionnaires to 20 people. Are they asked to review the questionnaire, or are they asked to answer it and then the results are analyzed (since the researchers can use the r-value to determine a valid question) ? or both? To explain "the required adjustment" referred to by the researcher.
2. The Author needs to explain more about the method used to ensure that respondents represent various regions of Saudi Arabia. Has the region been determined at the beginning of the study? How was the region selected? Is it achieved according to the target? It would be better if the name of the region were mentioned.
3. Time of questionnaire distribution needs to be added (e.g., in early 2019)
d. Methods need more detailed information to replicate. The Author can explain in more detail how to analyze the data. Researchers need to describe the conditions that must be met to use the test (Chi-square) and its alternatives.

Validity of the findings

a. The study has a meaningful impact.
b. All underlying data have been provided; they are robust, statistically sound, and controlled.
c. Conclusions are well-written

Additional comments

Dear Authors,
Congratulations on your work!
The author has tried to make significant improvements, but some improvements still can be made. The author can use general guidance reports like STROBE for cross-sectional study to make general improvement

Reviewer 3 ·

Basic reporting

1. The revisions regarding terminology consistency, title modification, paragraph reorganization, inclusion of current practices and policies in Saudi Arabia, and incorporation of recent literature are appropriate.
2. The elaboration on the hazards of unsafe medical disposal is also appropriate; however, the second sentence in the first paragraph, which begins with "Inappropriate pharmaceutical waste" (lines 78–79), should be removed as it does not fit within the paragraph and is already addressed in lines 94–95.
3. The manuscript structure adheres to standards, and the replacement of Table 3 with a figure while retaining Table 6 is reasonable.
4. An error remains in line 95 (punctuation before citation); thus, a thorough check for any remaining typographical errors is still necessary.
5. If table references cannot be placed inside parentheses, consider adding sentences for clarity, e.g., "The detailed participant characteristics can be found in Table 1."

Experimental design

1. The revisions ensuring consistency in study aims between the abstract and introduction, terminology use, and restructuring of study objectives are appropriate.
2. The elaboration on data privacy and confidentiality, as well as the addition of sample size calculation, references for questionnaire development, and supporting literature for the pilot study, are well addressed.
3. The inclusion of inferential statistical methods in the methodology section and the addition of Table 2 enhance clarity. However, to ensure consistency, tables presenting similar analyses should follow the same format (e.g., the format of Table 3 differs from Tables 4 and 5).
4. The inclusion of validity and reliability information, details on questionnaire domains, and statistical analysis methods in the abstract enhances clarity.
5. It is advisable to explicitly add the information regarding the prevention of multiple responses to the manuscript.

Validity of the findings

1. The revisions regarding sample size presentation, restructuring of the objectives, and improved logical flow in the results and discussion sections are appropriate.
2. The manuscript now reports only the final number of responses, improving clarity.
3. The discussion section was restructured as recommended, with necessary modifications to statements requiring clarification and referencing.
4. The addition of a limitations section, as well as recommendations for future research and practice, enhances the manuscript’s comprehensiveness.
5. The conclusion has been revised to align with the study’s objectives. However, the associated factors should be explicitly stated in the first part of both the discussion and conclusion sections, as they are currently not clearly highlighted despite being mentioned in many parts of the results and discussion.
6. Additionally, the abstract’s conclusion should be revised to explicitly address the research question by clearly stating the findings related to the level of awareness and associated factors.

Additional comments

The authors have made substantial improvements in clarity, structure, and methodological rigor. To further refine the manuscript, the associated factors should be explicitly summarized in the discussion and conclusion, and the abstract’s conclusion should clearly state the key findings. Overall, the revisions are well-executed, and with these final refinements, the manuscript will be even more polished. Well done!

---

## Round 0.3 · Minor Revisions

I appreciate the authors for their efforts in enhancing the content; nonetheless, additional enhancements are required in clarity, consistency, and alignment throughout the sections. The modifications implemented will improve clarity and effectiveness. I am confident that the authors' diligent efforts will significantly enhance the field.

·

Basic reporting

1. The improvement plan (line 53 and some other parts of the manuscript) should be "recommended improvement plan."
2. Tablet in the keyword is not directly related to research; maybe it can be replaced with other terms more related to research, such as "antibiotics."
3. The Author wrote a well-structured manuscript.
4. Figures are good
5. Raw data is well supplied.

Experimental design

1. The manuscript meets the original primary research requirement within the journal's Scope.
2. Overall, the research question is well-defined, relevant, and meaningful. It is stated how the research fills an identified knowledge gap.
3. Generally, research shows rigorous investigation performed with sufficient technical and ethical standards, but some improvements still can be made in some aspects, including :
4. Study tool. The questionnaire domain does not match the research objectives. The questionnaire domain indicated that the researcher would also collect data on "(recommended) improvement plan for safe disposal of used/expired medications," while the research objectives (lines 45-47; lines 133-135) indicated that the research was not directly related to such data.
The statement "to check the questionnaire" (line 175) needs to be added with a detailed explanation of whether the respondent is asked to fill in the questionnaire and provide suggestions for improvement or just fill in the questionnaire.
With pilot study data, researchers can consider the quality of the questionnaire only from the consistency of respondents' answers. However, in addition to measuring the consistency of respondents' answers, the quality of the questionnaire can also be improved by respondents' feedback. Did the researcher do both or only one of them?
5. Researchers should provide data on the Scope of distribution of the questionnaire (covering any region out of all regions) to ensure that the data obtained is representative of "all regions of the Kingdom of Saudi Arabia" (line 187).

Validity of the findings

1. The study has a meaningful impact.
2. All underlying data have been provided; they are robust, statistically sound, and controlled.
3. Conclusions are well-written

Additional comments

Dear Authors,
The author's efforts to make improvements are commendable, but there is still room for improvement.

Reviewer 3 ·

Basic reporting

Thank you for addressing the feedback on improving the clarity of the explanation on the hazards of unsafe medical disposal. The revisions regarding table references have also enhanced clarity. However, I still have concerns about basic reporting that may have been overlooked in the previous review:
1. Formatting Consistency
I appreciate the authors’ thorough check for typographical errors. However, please ensure consistency in subheading formatting. For example:
• Availability and disposal of unused/expired medications at home:
• Association between demographic characteristics and the presence and disposal of unused/expired medicines at home.
Some subheadings use a colon (:), while others use a period (.). For consistency, please follow a uniform style or omit punctuation, which is the more common practice.

2. Clarity, Grammar, and Language Quality
a. Clarity
• Lines 189-191: The sentence is repetitive. Please rephrase for conciseness.
• Lines 211-212: Consider simplifying to: All data were coded and analyzed using the Statistical Package for Social Sciences or SPSS version 22.0 (IBM Corp., Armonk, NY, USA).
• Lines 271-273: The sentence is complex; simplify for better clarity.
• Lines 289-291: Threaten the background is unclear—please clarify.
• Lines 333-336: Remove "and the current study results are in line with them" for better readability.
• Lines 336-339: The sentence is wordy—please rephrase concisely.
• Lines 339-340: Only at the required level is vague. Specify what required level means.
• Lines 341-342: Ensure significant findings are discussed first.
• Lines 388-391: Consider making the sentence more concise.
• Line 393: Instead of agreed/acknowledged/aware, explicitly state the statistical significance.
• Lines 403-407: The sentence is lengthy—consider splitting for readability.
• Lines 409-410: Clarify this study—does it refer to Ling et al., 2024?
b. Grammar
• Lines 264-265: The Neither… nor… nor… structure is awkward—please revise.
• Lines 305-306: Avoid starting a sentence with a number; restructure accordingly.
c. Redundancy
• Lines 147-148 and 187: The sentence "The study was conducted between the periods of January 2021 to December 2021" is repeated. Please vary the phrasing.
• Lines 174-178: Pilot study respondents is repeated multiple times—condense for clarity.
d. Past-present tense consistency
• Lines 251-256: The section inconsistently shifts between past and present tense. Ensure consistency.
• Lines 333-336: Shall contribute suggests a future implication while the rest is in past tense. Please revise.
• Lines 336-339: The sentence includes "It was suggested" (past) but later shifts to "be prescribed" (present). Consider maintaining a uniform tense.
• Lines 388-391: Ensure tense consistency when discussing current findings vs. previous studies.
e. Word choice
• Lines 305-306: Restructure the sentence to avoid starting with a percentage.
• Lines 336-339: “Encouraged to complete the given amount of the drugs” could be made more natural and direct.

3. P-value Reporting
I appreciate the authors’ efforts in revising the P-value reporting of P = 0.000 based on the previous reviewer’s suggestion. However, in the manuscript and Tables 3, 4, and 5, the revised values are reported as P < 0.01 instead of P < 0.001. For accuracy and consistency, please update all tables and text accordingly.

Experimental design

The authors have adequately addressed my previous comments on the experimental design. However, I realize that I previously overlooked the alignment between the study aim and the analysis performed, and I would like to highlight this now.
1. The study evaluates not only perceptions but also presence and disposal. To improve clarity, I suggest aligning the study objectives with the analysis performed to maintain focus throughout the manuscript.

2. This misalignment is also reflected in the Introduction, where you state:
"Therefore, the current study was undertaken to evaluate the general public’s perceptions regarding domestically disposing of unused/expired medicines. Additionally, it identifies the association between demographic characteristics and perceptions of safe disposal of unused/expired medicines. With these objectives, this study will help generate evidence regarding drug usage and disposal patterns among the general public living in Saudi Arabia."
However, "drug usage and disposal patterns" are not clearly reflected in the study objectives. These aspects extend beyond perception alone. To improve coherence, explicitly state all objectives and ensure they align with the order of the Results and Discussion.

Validity of the findings

I appreciate the revisions made in this section; however, further refinements are still needed to enhance clarity, consistency, and alignment with the study objectives.
1. Structural Inconsistency Between Results and Discussion
• The Results section presents Presence and Disposal Practices separately, while the Discussion combines them. This difference may confuse readers. Consider using a consistent structure in both sections.
• The Results mention Suggested Disposal Methods, to find in the Discussion. If preferred, you can place it under the Disposal Practices section, but ensure the structure remains consistent between the Results and Discussion.

2. Structuring the Discussion Section
• The Discussion should explicitly introduce the main study objectives at the beginning. This will help readers grasp the key findings before diving into detailed discussions.
• Follow a logical structure based on the study objectives, for example:
- Presence and associated factors
- Disposal and associated factors
- Perceptions and associated factors
• Associated factors, though considered secondary, are key findings and should be highlighted early in the Discussion.

3. Strengthening the Conclusion
• The Conclusion should explicitly summarize the three main aspects (presence, disposal, perception) and their associated factors, along with their implications.
• Currently, it is too general and does not clearly address the main objectives. Strengthening the Conclusion will:
- Enhance clarity and relevance by summarizing key takeaways in a structured manner.
- Strengthen the study’s contribution by clarifying how the findings advance knowledge.
- Guide future research and policy development by emphasizing practical implications.
• Since many readers rely on the Conclusion for key insights, a structured and comprehensive summary is essential for readability and impact.

4. Abstract Improvement
• The Results should explicitly report the three main findings and their associated factors:
- Presence of unused/expired medicines and associated factors
- Disposal practices and associated factors
- Perceptions and associated factors
• The Results should be data-driven, providing key statistics rather than general descriptions. Including quantitative measures will make the abstract more informative, precise, and impactful.
• Adjust the Conclusion to directly answer the research question, ensuring consistency between the Results and Conclusion so that the findings justify the recommendations.

Additional comments

I appreciate the authors' efforts in refining the manuscript; however, further improvements are needed in clarity, consistency, and alignment across sections. The revisions made will enhance readability and impact. I believe the authors' hard work will make a valuable contribution to the field.

---

## Round 0.4 · accepted · Accept

This revised version is suitable for publication in PeerJ.